# End-to-end dental pathology detection in 3D cone-beam computed tomography images

**Adel Zakirov**
Diagnocat
St. Petersburg, Russia
az@diagnocat.com

**Matvey Ezhov**
Diagnocat
Moscow, Russia
matvey@diagnocat.com

**Maxim Gusarev**
Diagnocat
Odessa, Ukraine
m.gusarev@diagnocat.com

**Vladimir Alexandrovsky**
Diagnocat
Moscow, Russia
va@diagnocat.com

**Evgeny Shumilov**
Diagnocat
Moscow, Russia
e.shumilov@diagnocat.com

## Abstract

Cone-beam computed tomography (CBCT) is a valuable imaging method in dental diagnostics that provides information not available in traditional 2D imaging. However, interpretation of CBCT images is a time-consuming process that requires a physician to work with complicated software. In this work we propose an automated pipeline composed of several deep convolutional neural networks and algorithmic heuristics. Our task is two-fold: a) find locations of each present tooth inside a 3D image volume, and b) detect several common tooth conditions in each tooth. The proposed system achieves 96.3% accuracy in tooth localization and an average of 0.94 ROC AUC for 6 common tooth conditions.

## 1  Introduction

Dental radiography plays an important role in disease detection and treatment planning. Dental images enable the dental professional to identify many conditions that may otherwise go undetected and to see conditions that cannot be identified clinically. It is probably the most frequent diagnostics in radiology - American Dental Association (ADA) recommends to recall Patients with No Clinical Caries and No Increased Risk for Caries every twelve months [1]. In 1999, a technology termed cone-beam computed tomography (CBCT) was introduced that allows for the viewing of structures in the oral-maxillofacial complex in three dimensions. CBCT has become a desired technology because of the accurate and detailed information it provides. At this time the indications for CBCT examinations are not well developed, but they are deemed dominantly useful in diagnosis and treatment planning of implant dentistry, endodontics, ENT, maxillofacial surgeries and others. Disadvantages of CBCT include the following:

- Time consumption and complexity for personnel to become fully acquainted with the imaging software and correctly using DICOM data. The ADA suggests that the CBCT image be evaluated by a dentist with appropriate training and education in CBCT interpretation.

- Many dental professionals who incorporate this technology into their practices have not had the training required to interpret data on anatomic areas beyond the maxilla and the mandible [7].

In recent years, deep learning has been successfully applied to various medical imaging problems, but its use remains limited within the field of dental radiography. Most applications work with 2D

1st Conference on Medical Imaging with Deep Learning (MIDL 2018), Amsterdam, The Netherlands.

X-ray images. In this paper, we present models for tooth localization and further tooth classification. The first model determines the locations of all the teeth. The second model focuses on a single tooth and detects several common tooth conditions: fillings, artificial crowns, implants, filled canals, and missing teeth.

## 2   Related work

Nowadays deep learning has radically transformed the medical imaging landscape, transitioning from scientific research labs to clinical applications [10]. Convolutional neural network architectures such as U-Net [20] for 2D and V-Net [14] and 3D U-Net [26] for 3D images have proven effective for anatomic structure segmentation on medical images, as well as [25, 17, 3]. Generative Adversarial Networks [4] are applied for medical image synthesis [15] and domain adaptation [9, 21].

Traditional computer vision approaches have seen some applications in dental imaging. [24] proposed a region-growing method for a tooth shape reconstruction, [23] used Radon transform and a combination of threshold and watershed methods to extract individual teeth regions, [19] proposed template matching for teeth segmentation. In [5] image processing techniques are used to create a 2D panoramic image from a 3D study tensor.

To our knowledge, applications of deep learning to dental imaging have been relatively sparse. [22] evaluates different deep learning methods for dental X-rays analysis. In [12, 13] authors used deep convolutional neural networks for tooth type classification and labeling on dental CBCT images. [2, 11] apply deep neural networks to caries detection. U-Net architecture was trained to segment dental X-ray images in the Grand Challenge for Computer-Automated Detection of Caries in Bitewing and won by a large margin.[8] presents an overview of machine and deep learning techniques in dental image analysis.

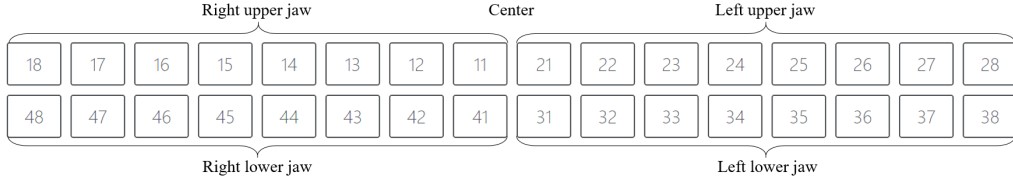

Figure 1: Tooth chart used in both localization and classification annotation.

## 3   Dataset

In this work, we used a dataset of depersonalized CBCT images. Patient consent was obtained for all participants in each study in the experiment. Images typically capture the lower region of a subject's face. Each dataset consists of images with different pixel spacing (from $0.15$ mm to $0.4$ mm) and field of view. A typical image comprises a stack of $432$ axial slices, each of size of $600 \times 600$ pixels. We believe that this dataset is representative for the majority of dental CBCT images obtained "in the wild". For annotation purposes, we used the European tooth chart in Figure 1. We annotated two separate datasets for each of the subtasks.

**Localization dataset**   A set of $517$ studies for localization was annotated by $4$ specialists. The annotation process consisted of selecting an axial slice, drawing a bounding box around the tooth axial profile, and entering a tooth number. The procedure for choosing axial slices to annotate varied during the period of annotation; typically we prepared a set of candidate slices consisting of an equally spaced subset from the full image, and the annotator chose $10 - 20$ slices from this subset. We obtained a dense 3D volumetric segmentation mask from sparse 2D annotations by linearly interpolating each bounding box corner to get a rough 3D shape. Then we took the superposition of pixel intensities and their center-distance's energy function.

The full procedure is described in section 4.1. The annotated teeth numbers distribution in our dataset is shown in Figure 2.

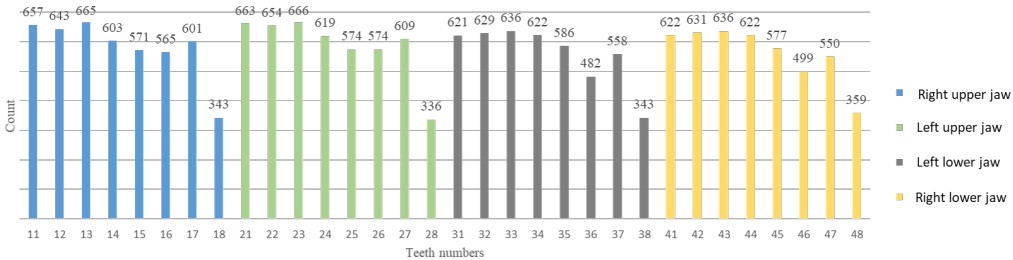

Figure 2: Teeth annotation distribution.

**Pathology dataset**   We have chosen a set of visually distinct tooth conditions: fillings, artificial crowns, implants, impacted tooth, filled canals, and removed or missing teeth. All of these conditions are visually easy to identify, although it takes a lot of time to describe all 32 teeth. A team of 5 specialists annotated a set of 1274 studies. The annotation process consisted of a specialist opening an anonymized study in a CBCT viewer of choice, inspecting the 3D image, and filling out a teeth chart by identifying conditions on each of the 32 teeth. Due to per-tooth example creation, 1274 annotated studies resulted in 39888 examples. A distribution of the conditions is presented in Figure 3.

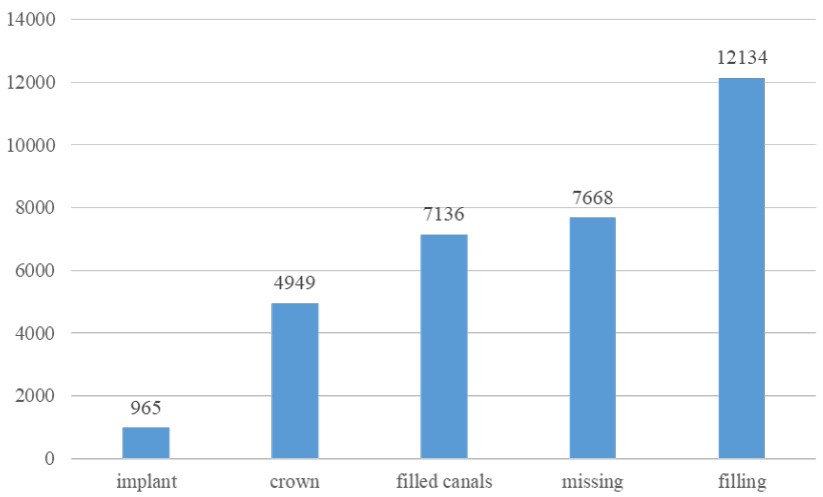

Figure 3: Tooth condition distribution.

## 4   Methods

Our approach consists of the following steps:

1. Preprocessing the incoming volumetric image.

2. Finding the location of each present tooth inside an image volume and identifying it by number (localization).

3. Extracting the tooth volumetric image together with the surrounding context.

4. Predicting a tooth's conditions based on its volumetric image (classification).

Steps 2 and 4 are implemented as separate deep convolutional neural networks. Steps 1 and 3 are algorithmic procedures.

## 4.1 Preprocessing

CBCT studies are received in the form of DICOM files. Files contain a 3D pixel array along with study metadata. Typically CT DICOM metadata contains instructions on how to transform the raw pixel array into an array of Hounsfield Unit (HU) radio intensity measurements. In case of CBCT, HU should be considered as pseudo-HU, since it is not certain that CBCT scanners can acquire accurate radio intensity measurements uniformly across the whole scanning volume [16]. We assume that this particularity should not be very important for analysis of CBCT images based on machine learning.

We preprocess images in several steps. Firstly, we clip the intensities to be inside the $0.05$ to $0.995$ quantile range to suppress outliers — CBCT scanner reconstruction errors. Then, for the localization model, we rescale the whole image to have $1.0^3$mm spatial resolution using linear interpolation. From the typical source resolution of 0.2mm, it translates into a 25x downscaling. For the classification model, we rescale the tooth sub-volume to have a fixed $64^3$ size with linear interpolation.

We do not change the underlying HU representation relying on model normalization layers. Before this, we tried different normalization schemes in an attempt to correct possible differences of intensity acquisition between devices. We found out that different normalization schemes either did not affect the final result, or had questionable trade-offs. We tried:

1. Using raw HU after clipping outliers: baseline.
2. Standardization and centering: no measurable difference.
3. Clipping below 0 and above $4000$ HU, then rescaling to the range $(0, 1)$: no measurable difference.
4. Global histogram equalization: significantly increases convergence speed, but results in slightly higher loss.
5. Localized energy-based normalization [18]: no measurable difference.

We decided to use raw HU because it simplifies reasoning about the process and improves interoperability between models in the pipeline.

**Segmentation masks preparation**   Manually annotated axial masks are sparse. Moreover, they are collected in the form of axial bounding boxes, which is non-standard for segmentation. To create a single continuous mask for each tooth, we developed an energy-based masks interpolation. Firstly, we create 33 masks (one for each tooth and one for the background) —- we measure mean distance from each pixel to the center of bounding boxes corresponding to each tooth. The closer the pixel is to the tooth's centerline, the more energy it has. The background also has energy, which is a unique trainable parameter. Secondly, we perform a linear combination of voxel intensities and voxel energies as:

$$energy\_intensities = intensities + k * energies,$$

where $k$ is a parameter. We have 33 energy masks afterwards and use the argmax function to label each voxel with a number from 0 to 32, where 0 is the background.

## 4.2 Localization

**Problem**   We formulate the problem of tooth localization as a 33-class semantic segmentation. Therefore, each of the 32 teeth and the background are interpreted as separate classes. We leave for future research the reformulation of the problem as instance segmentation.

**Model**   We use a V-Net [14]-based fully convolutional network. Our V-Net is 6 levels deep, with widths of $32, 64, 128, 256, 512, and 1024$. The final layer has an output width of 33, interpreted as a softmax distribution over each voxel, assigning it to either the background or one of 32 teeth. Each block contains $3 \times 3 \times 3$ convolutions with padding of 1 and stride of 1, followed by ReLU non-linear activations and a dropout with $0.1$ rate. We use instance normalization before each convolution. Batch normalization was not suitable in our case, as long as we had only one example in batch (GPU memory limits); therefore, batch statistics are not determined.

We tried different architecture modifications during the research stage. For example, an architecture with $64, 64, 128, 128, 256, 256$ units per layer leads to the vanishing gradient flow and, thus, no

training. On the other hand, reducing architecture layers to the first three (three down and three up) gives a comparable result to the proposed model, though the final loss remains higher.

**Loss function**  Let R be the ground truth segmentation with voxel values $r_i$ (0 or 1 for each class), and P the predicted probabilistic map for each class with voxel values $p_i$. As a loss function we use soft negative multi-class Jaccard similarity, that can be defined as:

$$JaccardMulticlassLoss = 1 - \frac{1}{N} \sum_{i=0}^{N} \frac{p_i r_i + \epsilon}{p_i + r_i - p_i r_i + \epsilon},$$

where $N$ is the number of classes, which in our case is 32, and $\epsilon$ is a loss function stability coefficient that helps to avoid a numerical issue of dividing by zero. Then we train the model to convergence using an Adam optimizer with learning rate of $1e-4$ and weight decay $1e-8$. We used a batch size of 1 due to the large memory requirements of using volumetric data and models. We stop training after 200 epochs and use the latest checkpoint (validation loss does not increase after reaching the convergence plateau).

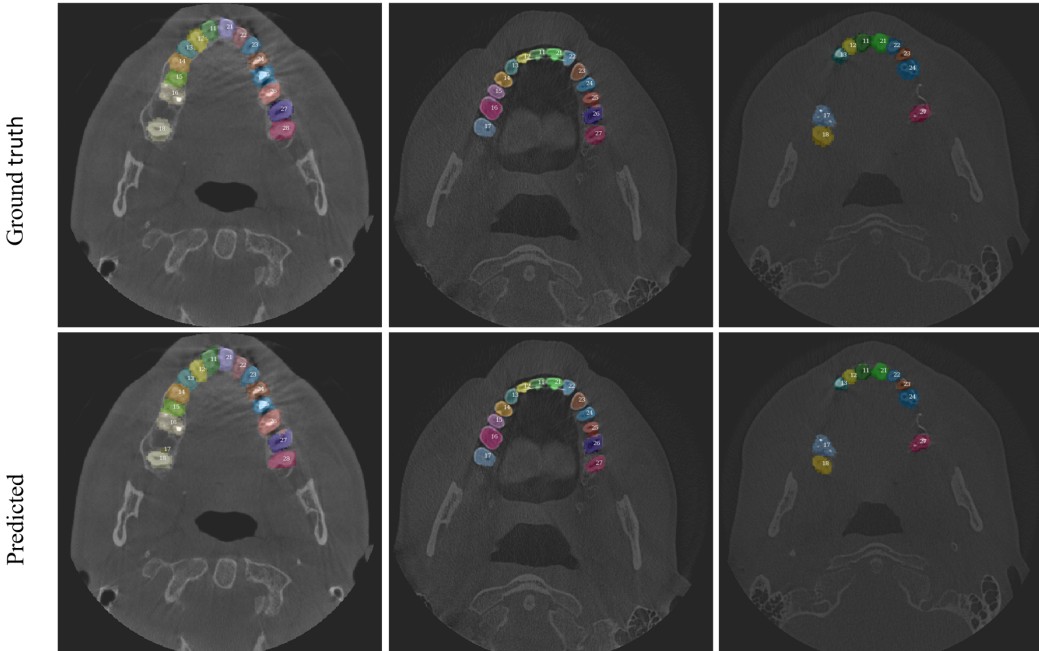

Figure 4: Examples of ground truth and predicted masks. Axial slices of original CBCT scans.

**Results**  The localization model is able to achieve a loss value of 0.28 on a test set. The background class loss is 0.0027, which means the model is a capable 2-way "tooth / not a tooth" segmentor.

We also define the localization intersection over union (IoU) between the tooth's ground truth volumetric bounding box and the model-predicted bounding box. In the case where a tooth is missing from ground truth and the model predicted any positive voxels (i.e. the ground truth bounding box is not defined), localization IoU is set to 0. In the case where a tooth is missing from ground truth and the model did not predict any positive voxels for it, localization IoU is set to 1.

For a human-interpretable metric, we use tooth localization accuracy which is a percent of teeth that have a localization IoU greater than 0.3 by our definition. The relatively low threshold value of 0.3 was decided from the manual observation that even low localization IoU values are enough to approximately localize teeth for the downstream processing. The localization model achieved a value of 0.963 IoU metric on the test set, which, on average, equates to the incorrect localization of 1 of 32 teeth. Figure 4 shows examples of teeth segmentation at axial slices of 3D tensor.

## 4.3 Tooth sub-volume extraction

In order to focus the downstream classification model on describing a specific tooth of interest, we extract the tooth and its surroundings from the original study as a rectangular volumetric region, centered on the tooth. In order to get the coordinates of the tooth, we rely on the upstream segmentation mask. The predicted volumetric binary mask of each tooth is preprocessed by applying erosion, dilation, and then selecting the largest connected component. We find a minimum bounding rectangle around the predicted volumetric mask. Then we extend the bounding box by $15$ mm vertically and $8$ mm horizontally (equally in all directions) to capture the tooth context and to correct possibly weak localizer performance. Finally, we extract a corresponding sub-volume from the original clipped image, rescale it to $64^3$ and pass it on to the classifier. An example of a sub-volume bounding box is presented in Figure 5.

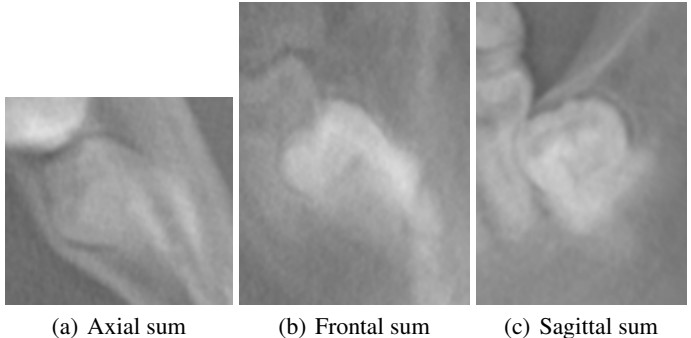

| (a) Axial sum | (b) Frontal sum | (c) Sagittal sum |

Figure 5: Example of extraction site produced by localization model. Three projections of a volumetric bounding box.

## 4.4 Classification

**Model**    The classification model has a DenseNet [6] architecture. The only difference between the original and our implementation of DenseNet is a replacement of the 2D convolution layers with 3D ones. We use $4$ dense blocks of $6$ layers each, with a growth rate of $48$, and a compression factor of $0.5$. After passing the $64^3$ input through $4$ dense blocks followed by down-sampling transitions, the resulting feature map is $548 \times 2 \times 2 \times 2$. This feature map is flattened and passed through a final linear layer that outputs $6$ logits — each for a type of abnormality.

**Loss function**    Since tooth conditions are not mutually exclusive, we use binary cross entropy as a loss. To handle class imbalance, we weight each condition loss inversely proportional to its frequency (positive rate) in the training set. Suppose that $F_i$ is the frequency of condition $i$, $p_i$ is its predicted probability (sigmoid on output of network) and $t_i$ is ground truth. Then:

$$L_i = (1/F_i) \cdot t_i \cdot \log p_i + F_i \cdot (1 - t_i) \cdot \log(1 - p_i)$$

is the loss function for condition $i$. The final example loss is taken as an average of the 6 condition losses.

**Results**    The classification model achieved average ROC AUC of $0.94$ across the 6 conditions. Per-condition scores are presented in Table 1. ROC curves of the 6 predicted conditions are illustrated in Figure 6.

Table 1: ROC AUC metrics and % of positive cases in train set

|  | Artificial crowns | Filling canals | Filling | Impacted tooth | Implant | Missing |
|---|---|---|---|---|---|---|
| ROC AUC | 0.941 | 0.95 | 0.892 | 0.931 | 0.979 | 0.946 |
| Condition frequency | 0.092 | 0.129 | 0.215 | 0.018 | 0.015 | 0.145 |

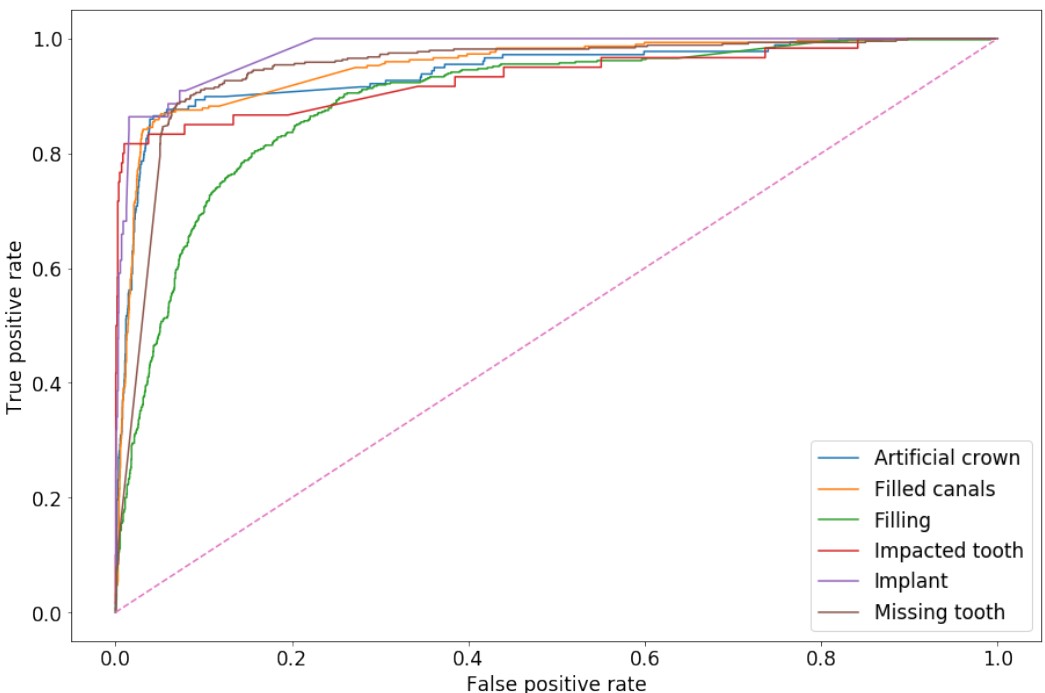

Figure 6: Resulting ROC curves for a classification task

# 5 Future work

Further research and development is needed to improve the performance of the pipeline. This includes model improvements as well improvements to the data. Our plans include:

- Increasing the quality of data by asking a committee to annotate the same studies.
- Increasing the size of the datasets, especially required for the classification task.
- Adding volumetric data augmentations during training.
- Reformulating the localization task as instance segmentation instead of semantic segmentation. All the teeth are visually very similar, so treating them as separate classes might decrease the ability of a model to generalize.
- Reformulating the localization task as object detection. Object detection might be more suited for our case, as the end-goal is to find and extract the general location of a tooth for further processing.
- Attempting different class imbalance handling approaches for the classification model.
- Localizing and extracting the jaw region of interest as a first step in the pipeline. The jaw region typically takes around 30% of the image volume and has adequate visual distinction. Extracting it with a shallow/small model would allow for larger downstream models.
- Expanding diagnostic coverage from 6 basic tooth conditions to other diagnostically relevant conditions and pathologies.

# 6 Conclusion

In this work, we developed an end-to-end pipeline for detecting 6 common teeth states in dental 3D CBCT scans. Our approach consists of localizing each present tooth inside an image volume and predicting tooth states from the volumetric image of a tooth and its surroundings. We collected and annotated datasets of 517 and 1274 3D images for localization and classification tasks respectively. Our V-Net based localization model shows a localization accuracy value of 0.96. Our DenseNet

based model ROC achieves an average AUC value of $0.94$ on the teeth states binary classification task. The classification performance achieved is not yet high enough for use in a clinical setting as a diagnostics assistant. At the same time, the performance of the localization model allows us to build a high-quality 2D panoramic reconstruction, which provides a familiar and convenient way for a dentist to inspect a 3D CBCT image. Methods used to create panoramic reconstructions will be described in future works.

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
