# OpenReview forum: "End-to-end dental pathology detection in 3D cone-beam computed tomography images"
_MIDL.amsterdam/2018/Conference — Submitted to MIDL 2018_

### Review · AnonReviewer2 · 2018-05-07
**Good work, but needs to show more results**

**Rating:** 2
**Confidence:** 2

**Review:**

This work explores the utilization of deep learning for the localization of teeth in 3D CT scans, as well as the classification of several commonly-found conditions. The tooth localization task is formulated as a 33-class semantic segmentation (32 teeth + background), and uses a V-Net-based, fully-convolutional network. Classification is performed on a per-tooth basis (based on the results of localization) using a DenseNet with 3D layers. Results seem good, but more explanation and validation is required.

Here are some specific notes:
- No training/testing/validation splits were mentioned for either localization or classification
- Some minor typos and grammatical errors
- No report of mean IoU for localization -- this makes it difficult to assess and interpret performance
- Would have been good to see what different IoU's look like to ensure that 0.3 is actually an acceptable threshold, as the standard is usually around 0.7.  Without this, the accuracy rate of 96.3% is difficult to rationalize and interpret. A figure depicting the visual results of different IoU's and maybe a plot of the distribution IoU values would be beneficial here.
- Potential further analysis: Analyze localization IoU as a function of tooth number/location. Might yield some insightful results.
- For classification, are all images (3D) input in their existing orientation? Does flipping and rotating the images at training time change performance? This could make the model direction/rotation-invariant, likely yielding better results.
- Figure 3: Only shows 5/6 conditions analyzed (missing "Impacted")
- For classification, were all teeth classified as a "pathology"? Were no "normal" teeth included? I feel like this label should also be included if planning to be used in the field.
- Table 1: "% of positive cases in train set" -- if all the %'s are added up, they don't equal 1.0. Something is missing here.
- Classification results: the inclusion of a confusion matrix would likely be beneficial to this work. Does the model biased to any labels? Are any labels commonly confused?
- If a section is labelled "Methods", it shouldn't contain "Results". The sections should either be labelled differently, or the results should all be moved to their own section.

Overall: interesting work, but requires more validation in order to be accepted.

**Special Issue:**

No

---

> ### Comment · ~Matvey_Ezhov1 · 2018-06-26
> **Thank you for the review**
>
> Dear reviewer, we thank you for the time spent reviewing our submission. We realize now that our treatment of the experiment setup was not as thorough as it should have been. We will comprehensively improve our paper and address this and other problems you highlighted before submitting again.
> To obtain the reported results we used train/dev/test split with approximately 7% of all examples going into dev and test sets each. Each study was present in either localization or classification datasets but not in both.
> Since conditions are not mutually exclusive, we decided to treat the problem as multiple binary classification problem. Data in table 1 “% of positive cases in train set” should not be summed, rather it is interpreted as “negative = 1-positive”. Also, it is not clear how we can build confusion matrix for this case, but we agree that additional exploratory data should be included in the paper.
> We did not use any data augmentation for this work.
> Thanks again for the review and improvement ideas!

---

### Review · AnonReviewer3 · 2018-05-08
**NOT end-to-end, testing setup not clear**

**Rating:** 2
**Confidence:** 2

**Review:**

Pro: The paper is easy to follow, the full pipeline from dental CT to tooth-wise classification is presented. Results on a fairly large data set are presented.

Con: I understand there may be different perspectives to this, but to me this is _not_ "end-to-end CT" as it doesn't start with the raw projections. Moreover, there are several steps set up individually, they are not combined, optimized, and trained jointly. The results look promising, but it's not clear what the training and testing set up is - is it an independent test set? Are training errors presented? There are several steps - is it a cross-validation in every step (might over train), or a full evaluation of all steps?

**Special Issue:**

No

---

> ### Comment · ~Matvey_Ezhov1 · 2018-06-26
> **Thank you for the review**
>
> Dear reviewer, we thank you for the time spent reviewing our submission. We realize now that our treatment of the experiment setup was not as thorough as it should have been. We will comprehensively improve our paper and address this and other problems you highlighted before submitting again. Also, we will refrain from calling our method end-to-end.
> To obtain reported results we used train/dev/test split with approximately 7% of all examples going into dev and test sets each. Each study was present in either localization or classification datasets but not in both.

---

### Review · AnonReviewer1 · 2018-05-08
**The authors propose an end-to-end pipeline for detecting common teeth states using CBCT scans. This multi-step process employs standard deep learning algorithms to localize teeth positions, followed by classifying them into different states. Though well written, there are significant details missing regarding implementation strategies which need to be addressed.**

**Rating:** 1
**Confidence:** 2

**Review:**

The following issues need to be addressed comprehensively to improve the quality of work:

1. Abstract needs to be more specific regarding the steps of the pipeline and the algorithms used. Just stating 'several deep CNNs and algorithmic heuristics' is not informative enougt
2. Two disadvantages of CBCT have been enlisted in the introduction section. How do the authors address this with the developed pipeline, particularly the time consumption and complexity issues?
3. In the methods section, Steps 2 and 4 are stated as deep CNNs and steps 1 and 3 are algorithmic procedures. Authors should note that 2, and 4 are also algorithms. Just because they are CNNs doesn't mean that they are not algorithms
4. How was the global intensity equalization performed? Was this an adaptive process?
5. The major drawback of the work is the unavailability of any data splitting information and a highly unclear description of the experimental design. Are the performance metrics reported on an independent discovery set, or was it cross validation? In any case, what was the N
6. Please include a dedicated Results section
7. How was the 15mm vertical and the 8mm horizontal extension decided upon?

**Special Issue:**

No

---

> ### Comment · ~Matvey_Ezhov1 · 2018-06-26
> **Thank you for the review**
>
> Dear reviewer, we thank you for the time spent reviewing our submission. We realize now that our treatment of the experiment setup was not as thorough as it should have been. We will comprehensively improve our paper and address this and other problems you highlighted before submitting again.
> Regarding 4, we tested standard global equalization - transforming image to have a flat histogram of intensity values. It was not adaptive in any sense beyond measuring CDF of image intensities.
> Regarding 5, to obtain reported results we used standard train/dev/test split with approximately 7% of all examples going into dev and test sets each.
> Regarding 7, it was decided by the expert opinion (radiologist estimating that condition-relevant context is no bigger than this).

---

### Decision · Program_Chairs · 2018-05-15
**Paper91 Acceptance Decision**

Reject